# Prediction of Fluid Responsiveness Using Combined End-Expiratory and End-Inspiratory Occlusion Tests in Cardiac Surgical Patients

**DOI:** 10.3390/jcm12072569

**Published:** 2023-03-29

**Authors:** Jan Horejsek, Martin Balík, Jan Kunstýř, Pavel Michálek, Tomáš Brožek, Petr Kopecký, Adam Fink, Petr Waldauf, Michal Pořízka

**Affiliations:** 1Department of Anaesthesiology and Intensive Care Medicine, First Faculty of Medicine, Charles University in Prague and General University Hospital in Prague, 12808 Prague, Czech Republic; 2Department of Anaesthesia, Antrim Area Hospital, Antrim BT41 2RL, UK; 3First Faculty of Medicine, Charles University in Prague, 12808 Prague, Czech Republic; 4Department of Anaesthesiology and Resuscitation, Third Faculty of Medicine, Charles University in Prague and University Hospital Královské Vinohrady in Prague, 10034 Prague, Czech Republic

**Keywords:** end-expiratory occlusion, end-inspiratory occlusion, fluid responsiveness, circulatory shock, hypovolemia, pulse contour analysis

## Abstract

End-expiratory occlusion (EEO) and end-inspiratory occlusion (EIO) tests have been successfully used to predict fluid responsiveness in various settings using calibrated pulse contour analysis and echocardiography. The aim of this study was to test if respiratory occlusion tests predicted fluid responsiveness reliably in cardiac surgical patients with protective ventilation. This single-centre, prospective study, included 57 ventilated patients after elective coronary artery bypass grafting who were indicated for fluid expansion. Baseline echocardiographic measurements were obtained and patients with significant cardiac pathology were excluded. Cardiac index (CI), stroke volume and stroke volume variation were recorded using uncalibrated pulse contour analysis at baseline, after performing EEO and EIO tests and after volume expansion (7 mL/kg of succinylated gelatin). Fluid responsiveness was defined as an increase in cardiac index by 15%. Neither EEO, EIO nor their combination predicted fluid responsiveness reliably in our study. After a combined EEO and EIO, a cut-off point for CI change of 16.7% predicted fluid responsiveness with a sensitivity of 61.8%, specificity of 69.6% and ROC AUC of 0.593. In elective cardiac surgical patients with protective ventilation, respiratory occlusion tests failed to predict fluid responsiveness using uncalibrated pulse contour analysis.

## 1. Introduction

Intravenous fluids are commonly administered in patients with acute circulatory failure to improve cardiac output (CO) and tissue perfusion. However, a significant CO increase is only observed in approximately 50% of patients, the so-called fluid responders [1,2]. Fluids profoundly impact clinical outcomes, including morbidity, mortality and length of hospital stay if administered insufficiently or in excess [3,4,5,6]. The unreliability of static parameters of fluid responsiveness, such as central venous pressure (CVP) [7] or pulmonary artery wedge pressure [8], has led to the development of dynamic tests which can differentiate between fluid responders and non-responders with higher precision [9]. Because of their respective limitations [9,10,11], clinicians need several dynamic tests to predict fluid responsiveness in various scenarios.

The end-expiratory occlusion (EEO) test is a dynamic test based on heart-lung interactions. Mechanical ventilation interruption at the end of expiration transiently decreases intrathoracic pressure producing an internal fluid challenge and a temporary increase in CO in fluid-responsive subjects [12]. Similar to EEO, the end-inspiratory occlusion (EIO) test interrupts ventilation at the end of inspiration, increasing intrathoracic pressure and decreasing cardiac preload, leading to lower CO. The addition of EEO and EIO changes has been used to increase the diagnostic threshold of CO changes [13,14]. Conflicting data exist regarding the discriminatory ability of these tests to predict fluid responsiveness accurately. Several studies, including one meta-analysis, report an excellent prediction of fluid responsiveness in critically ill patients, even those with arrhythmias and acute respiratory distress syndrome (ARDS) [12,13,14,15,16,17,18,19,20,21,22]. Respiratory occlusion tests may also perform better in patients with elevated intra-abdominal pressure where otherwise popular passive leg raising may produce false negative results [23]. On the contrary, other reports documented a failure of EEO to predict fluid responsiveness in medical and general surgical patients with protective mechanical ventilation [17,24]. Ventilation with low tidal volumes of 6–7 mL/kg and less is considered a significant limitation in the use of dynamic parameters, including the widely used pulse pressure variation (PPV) and stroke volume variation (SVV). Low tidal volumes induce only minor changes in intrathoracic pressure that may not produce an adequate hemodynamic response. Furthermore, this method has never been tested in patients in the early postoperative period after cardiac surgery, whose baseline preload status may differ from septic shock patients. Thus, our study aimed to evaluate the diagnostic accuracy of respiratory occlusion tests to predict fluid responsiveness in cardiac surgical patients with protective mechanical ventilation.

## 2. Materials and Methods

### 2.1. Patients

This prospective, single-centre study was approved by the Institutional Review Board (Ethics Committee of the General University Hospital, Prague, Czech Republic, No. 992/19 S-IV). Informed consent was obtained from all participants. Patients scheduled for elective coronary artery bypass grafting in the General University Hospital in Prague between October 2019 and July 2022 were screened for enrolment. We included sedated and mechanically ventilated patients over 18 years of age with normal systolic function of the left and right ventricle, defined as left ventricular ejection fraction (LVEF) > 50% and right ventricular fractional area change (RV FAC) > 30%, in whom volume expansion was indicated by the attending physician based on any of the signs of suspected hypovolemia if study investigators were available. These parameters included central venous oxygen saturation < 65%, arterial lactate concentration > 2 mmol/L, any vasopressor support with norepinephrine, CVP < 5 mmHg and the presence of skin mottling. The exclusion criteria included spontaneous breathing activity interfering with a 15-s respiratory occlusion, any arrhythmias, moderate to severe valvular heart disease, aggressive mechanical ventilation (defined as positive end-expiratory pressure (PEEP) > 10 cm H_2_O or peak inspiratory pressure > 30 cm H_2_O), any use of inotropes besides norepinephrine, poor echogenicity and an open thorax. Poor echogenicity was defined as the inability to visualise both ventricles and perform the required measurements.

### 2.2. Haemodynamic Monitoring

All patients were equipped with a radial artery catheter and a central venous catheter in the right internal jugular vein. Standard monitoring included 5-lead electrocardiography, invasive blood pressure, central venous pressure, pulse oximetry and heart rate (Solar 8000 M, GE Healthcare, Chicago, IL, USA). Cardiac output, cardiac index (CI), stroke volume (SV) and SVV were measured continuously using uncalibrated pressure waveform analysis (FloTrac^TM^/EV1000^TM^, Edwards Lifesciences, Irvine, CA, USA). 

Transthoracic echocardiographic measurements were performed using the GE Vivid S6 Ultrasound Machine (GE Healthcare, Chicago, IL, USA). From the apical 4-chamber view, we assessed LVEF and RV FAC. The LVEF was calculated using the monoplane Simpson’s method. All measurements were performed by two physicians (MP and MB) holding a certificate in echocardiography. Recordings were analysed offline by a single observer (MP) blinded to study data. 

### 2.3. Mechanical Ventilation

All patients were ventilated using the volume control/assist mode (Hamilton C1, Hamilton Medical, Bonaduz, Switzerland) with tidal volumes of 7 mL/kg of ideal body weight (IBW). The attending physician set the FiO_2_, PEEP and respiratory rate based on the initial arterial blood gas analysis. In the study population, the PEEP was kept between 4 and 6 cm H_2_O, the respiratory rate between 10 and 14 per minute and FiO_2_ between 0.4 and 0.5. None of these parameters were adjusted during the study period.

### 2.4. Study Design

Patients were transferred from the operating theatre at the end of surgery to the ICU and sedated by intravenous propofol infusion. At admission, mechanical ventilation was resumed using the settings mentioned above, baseline arterial and venous blood gas analyses were performed and a chest X-ray was taken. Mean arterial pressure was maintained between 65–80 mmHg with continuous infusion of norepinephrine. Patients were included in the study if the attending physician indicated volume expansion and sufficient echogenicity was confirmed. All patients were in the supine position with an elevated trunk of 15°. The first set of hemodynamic and echocardiographic measurements was taken at baseline. The second set was taken immediately before and the third after a 15-s EEO, performed as described in the original study by Monnet et al. [1]. The fourth and fifth sets were taken before and after a 15-s EIO, performed as described by Jozwiak et al. [2]. Both respiratory occlusion tests were separated by a time window of 1 minute to allow the hemodynamic parameters to return to baseline. The last set of measurements were taken after a fluid challenge (FC) using succinylated gelatin (Geloplasma, Fresenius Kabi, Prague, Czech Republic) at a dose of 7 mL/kg administered over 15 min. Patients were divided into two study groups (fluid responders and non-responders) based on fluid responsiveness, defined as an increase in CI ≥ 15%. The demographic and medical history data, hemodynamic parameters including heart rate (HR), mean arterial pressure (MAP), CVP, CI, SV and SVV were compared between the study groups at baseline, during respiratory occlusion tests and after fluid challenge as outlined above.

### 2.5. Statistical Analysis

R version 4.2.2 [25] with graphical user interface RStudio version 2022.07.2 [26] was used for the statistical analysis. Data wrangling and visualization were performed using a collection of libraries for data science Tidyverse version 1.3.2 [27]. Exploratory data analysis was performed for all parameters. The normality of data was assessed with the Kolmogorov-Smirnov test. Data were summarized as median (25–75% interquartile range) or mean (standard deviation, SD) as appropriate. A chi-square test, followed by a Fisher’s exact test as appropriate, was used for comparisons of categorical data. Differences between the groups were assessed using Student’s t-test for normally distributed variables and the Wilcoxon test for non-normally distributed variables. The ability of CI change, induced by individual respiratory tests, or their combination, to predict fluid responsiveness was evaluated using logistic regression followed by receiver operating characteristic (ROC) analysis. The Youden index (sensitivity + specificity − 1) [28] was used for the binarisation of continuous variables and the calculation of optimal cut-off points, library Cutpointr version 1.1.2 [29]. The diagnosis sensitivity, specificity and positive and negative predictive values were calculated for all parameters. A *p*-value of <0.05 was considered statistically significant.

## 3. Results

### 3.1. Patient Characteristics

During the study period (October 2019 to July 2022), 106 patients met all inclusion criteria. Forty-four (41.5%) were excluded due to poor echogenicity, in agreement with previous studies [30,31], three (2.8%) due to spontaneous breathing activity impeding the respiratory occlusion tests and two (1.9%) due to hemodynamic instability induced by arrhythmia (Figure 1). The characteristics of the included 57 patients are summarized in Table 1. Thirty-four patients (59.6%) were fluid responders. There were no significant differences in patient characteristics between the study groups, except for the difference in EuroSCORE II (0.89 [0.68–1.47] for responders versus 1.25 [0.91–1.62] for non-responders).

### 3.2. Hemodynamic Effects of EEO, EIO and Volume Expansion

At baseline, SVV was significantly higher in responders compared to non-responders (17.1 ± 7.9 vs. 12.9 ± 5.1, respectively, *p* < 0.05). All other hemodynamic parameters, including variations between individual baselines (baseline 1, 2 and 3), did not differ between the groups (Table 2).

Before volume expansion, there was a trend towards a larger magnitude of changes in CI and SV induced by EIO, EEO and their combination in responders compared to non-responders, including the added effects of EEO and EIO on cardiac index (19 ± 11 vs. 16 ± 10%, respectively, *p* = 0.4) and stroke volume change (20 ± 12 vs. 16 ± 10%, respectively, *p* = 0.2) (Figure 2). However, all of these differences failed to reach statistical significance in our study. Volume expansion resulted in a markedly higher increase in CI (26 ± 8 vs. 7 ± 4 %, respectively, *p* < 0.05) and SV (29 ± 13 vs. 11 ± 7%, respectively, *p* < 0.05) in responders than in non-responders. All changes in hemodynamic parameters induced by the respiratory tests and volume expansion are summarized in Table 3. The norepinephrine infusion rate was not changed in any of the patients during respiratory tests and volume expansion.

### 3.3. Prediction of Fluid Responsiveness

Neither EEO, EIO nor their combination predicted fluid responsiveness reliably in our study. Univariate logistic regression analysis was performed for continuous and binarized variables that were analysed as predictors of fluid responsiveness. These included CI and SV changes during respiratory occlusion tests (individual and combined) and baseline SVV, showing no significant predictive ability (Table 4). In ROC analysis, a cut-off point for CI change of 5.3% after EEO yielded a sensitivity of 70.6% and specificity of 60.9% with ROC AUC of 0.590. The discriminatory ability to predict fluid responsiveness was similar in the EIO test with a CI change cut-off point of—8.3%, sensitivity of 64.7%, specificity of 47.8% and ROC AUC of 0.552. After a combined EEO and EIO test, a cut-off point for CI change of 16.7% predicted fluid responsiveness with a sensitivity of 61.8% and a specificity of 69.6% and ROC AUC of 0.593. Similarly, a poor ability to predict fluid responsiveness in respiratory occlusion tests was also obtained for SV change (Table 5).

In the case of baseline SVV, the sensitivity and specificity were 58.8% and 69.6%, respectively, for a cut-off point of 14% with ROC AUC of 0.645. The ROC curves for the summary changes in CI and SV after an EEO and EIO as well as for baseline SVV are displayed in Figure 3. A summary of cut-off points, sensitivity, specificity and positive and negative predictive values is presented in Table 5.

## 4. Discussion

Our study did not confirm the previously reported accuracy of EEO for predicting fluid responsiveness in elective cardiac surgical patients in the ICU. Several explanations can be proposed for this discrepancy, as well as the limitations of this study. The EEO test has been increasingly used to predict fluid responsiveness with excellent results in the operating theatre [18,20] and the ICU [12,14,16,22]. Because the diagnostic threshold of EEO is close to the accuracy of routinely employed methods, especially echocardiography [13,32,33], Jozwiak et al. combined the changes induced by EEO and EIO, increasing the diagnostic threshold of CO change to 13% while keeping excellent sensitivity and specificity [13]. These results were replicated in a later study by Depret et al. [14]. In the case of EEO, a recent meta-analysis of 13 studies showed a pooled sensitivity of 0.85 and specificity of 0.88 with ROC AUC 0.91 at the threshold change in CO of 5.1% [15]; there was no significant difference between studies using pulse contour analysis or echocardiography to assess these changes and the test remained reliable at different PEEP settings and variable tidal volumes. 

Nevertheless, several studies also report a failure of respiratory occlusion tests to predict fluid responsiveness reliably. In a study by Guinot et al. [24], failure of EEO in predicting fluid responsiveness was noted in surgical patients who were ventilated with low PEEP and inspiratory plateau pressures intraoperatively. This finding is in agreement with our study. All our study patients were ventilated with a lung protective strategy, receiving tidal volumes of 7 mL/kg IBW and PEEP of 4–6 cm H_2_O, which may have decreased the magnitude of changes in venous return introduced by EEO and EIO [34]. Even though EEO has been reported to retain its reliability even in low tidal volume ventilation or low PEEP [15], only two studies combined both ventilatory settings [16,18]. There seems to be a threshold tidal volume where EEO loses its diagnostic value—in studies where tidal volume was set to 6 mL/kg, EEO could not predict fluid responsiveness accurately [17,20]. In general, protective mechanical ventilation represents a significant limitation in using dynamic indices of fluid responsiveness, especially pulse pressure and stroke volume variation. The use of low tidal volumes may cause only minor changes in intra-thoracic pressure which do not have to cause a significant hemodynamic change even in the presence of hypovolemia. Such an assumption is supported by our data as SVV underperformed in the prediction of fluid responsiveness (ROC AUC 0.645) (Figure 3). In contrast, another study by Myatra et al. [17] showed that the EEO test displayed a lack of discriminatory ability to predict fluid responsiveness in ICU patients with low respiratory system compliance below 30 mL/cm H_2_O ventilated with high PEEP levels and low tidal volumes. The presence of ARDS with low lung compliance can also limit the transition of pressure changes from the airways to the cardiovascular system during mechanical ventilation. On the other hand, the original study by Monnet et al. [12] demonstrated excellent discrimination of fluid responsiveness with the EEO test in ARDS patients with low tidal volume ventilation. However, the entire study population included patients in septic shock with presumed profound preload deficit. Our study population was comprised of elective surgical patients who were more likely resuscitated with fluids to a greater extent compared to septic shock patients. Thus, in such a clinical setting, respiratory occlusion tests may also not have to cause sufficient hemodynamic changes to detect fluid responsiveness, possibly even in patients with hypovolemia.

We used the uncalibrated pulse-contour analysis by the FloTrac^TM^/EV1000^TM^ system to calculate CO, SV and SVV without external calibration in our study. However, most studies to date have relied on calibrated pulse contour analysis [12,16,22] or echocardiography [13,19] for CO measurements, with only two using uncalibrated pulse contour analysis [18,20], but none in the ICU. This is the first study to date to evaluate the ability of FloTrac^TM^/EV1000^TM^ to assess the hemodynamic effects of EEO and EIO, as well as the first study to use an uncalibrated pulse contour analysis method in the ICU. Calibrated pulse contour analysis devices are considered more accurate than uncalibrated devices as their concordance with pulmonary artery catheter measurements is the highest [35], even though inter-device agreement has been questioned [33]. Calibrated systems are generally more expensive, necessitating either a special thermistor-tipped arterial catheter with a central venous line (PiCCO^TM^, VolumeView^TM^) or lithium injections with a lithium dilution sensor (LiDCO Plus^TM^), and require frequent recalibration to maintain their precision [36]. On the other hand, uncalibrated devices either do not need any disposable equipment (MostCare^TM^, LiDCO Rapid^TM^) or only rely on dedicated sensors (FloTrac^TM^, ProAQT^TM^), thereby cutting costs and avoiding complications associated with arterial catheter re-insertion in patients where circulatory instability had not been expected initially and a thermistor-tipped catheter was not used. Despite being less accurate, uncalibrated systems seem to have sufficient precision for CO measurement and are favored in situations where the circulation is normo- or hypodynamic [37] as well as in the first hours after cardiac surgery [38], although conflicting data have been reported, predominantly in patient groups where vascular tone changed significantly [39,40,41]. The FloTrac^TM^/EV1000^TM^ system has improved in its ability to adjust for changes over the years, especially in vascular tone. In our study, we used the latest, fourth generation software, which seems to have improved performance in scenarios with acute vascular resistance changes [42,43]. However, these advances may still not be sufficient to monitor patients receiving vasoactive agents with adequate precision [42,44]. Furthermore, the EEO-induced changes in hemodynamic parameters last only several seconds [45] while the FloTrac^TM^ algorithm displays CO, SV and SVV based on data averaged over 20-s periods [42,46]. Therefore, in this case, the uncalibrated analysis may not adequately detect such rapid and transient hemodynamic changes and could provide false negative results even in fluid-responsive patients. Thus, in light of the aforementioned technical and clinical limitations, our study confirms the previous findings of the limited utility of respiratory occlusion tests in predicting fluid responsiveness in the surgical population of protectively ventilated patients [17,20,24]. The present study has several limitations. The low number of selected patients and high patient study exclusion rate of 46% may represent a significant drawback. We performed only 15-s respiratory occlusion tests as described in the original studies. However, prolonging the apnoeic pause to 30 s could increase its sensitivity as shown in a recent study [18]. Furthermore, the patient’s enrolment in the study was decided by an attending clinician based on the presence of any clinical and laboratory signs of suspected hypovolemia and tissue hypoperfusion (Section 2). That could have led to the higher inclusion of sufficiently fluid-resuscitated patients at baseline, mainly when unreliable criteria, including CVP or administration of vasopressor support, were used alone. The major limitations regarding low tidal mechanical ventilations and technical issues of uncalibrated pulse contour analysis have already been discussed.

In conclusion, individual respiratory occlusion tests or their combination failed in predicting fluid responsiveness in elective, protectively ventilated, cardiac surgical patients. The technical limitations of uncalibrated pulse contour analysis may play a crucial role in detecting subtle cardiac output changes during the tests.

## Figures and Tables

**Figure 1 jcm-12-02569-f001:**
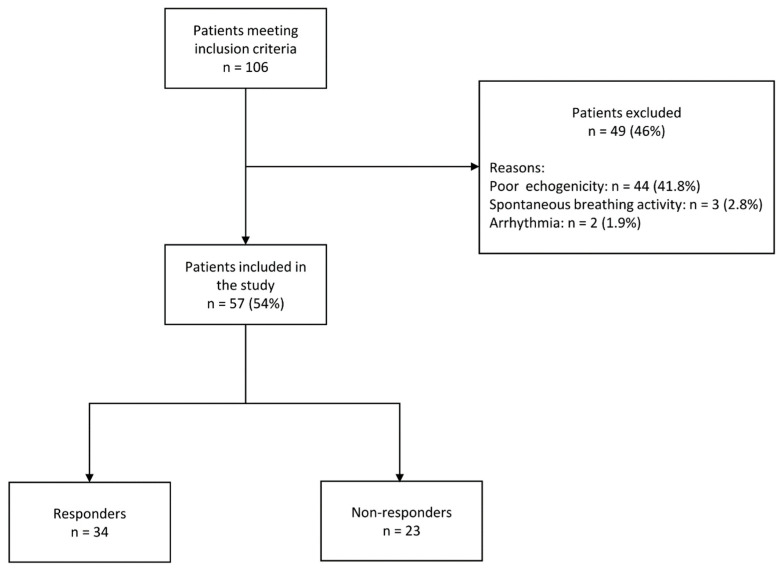
Flowchart of patient inclusion.

**Figure 2 jcm-12-02569-f002:**
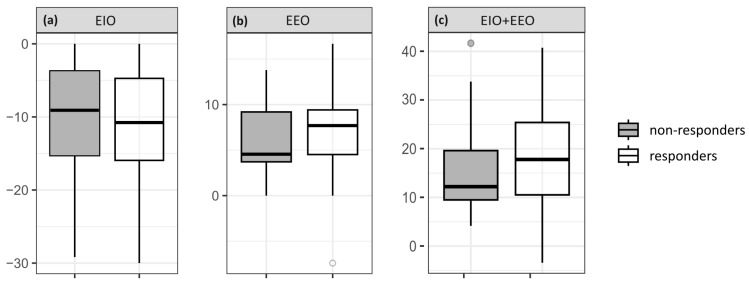
Boxplots displaying the changes in CI for (**a**) EIO, (**b**) EEO and (**c**) combined EIO + EEO.

**Figure 3 jcm-12-02569-f003:**
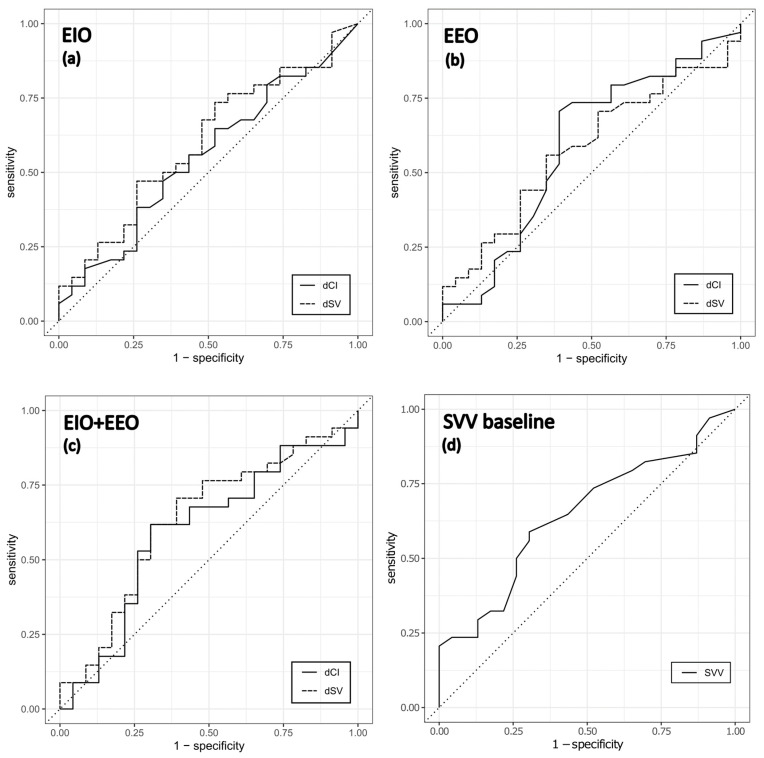
Receiver operating characteristic (ROC) curves evaluating the ability of changes in CI and SV to predict fluid responsiveness during (**a**) EIO, (**b**) EEO and (**c**) combined EIO and EEO maneuvers; (**d**) shows the ROC curve of fluid responsiveness prediction based on baseline SVV.

**Table 1 jcm-12-02569-t001:** Characteristics of the study population.

	Responders *n* = 34	Non-Responders *n* = 23	*p*-Value
Age (years)	65 ± 9	66 ± 8	0.3
Sex (male/female)	26/8	19/4	0.7
BMI	30.5 (26.4–32.7)	28.7 (25.7–32.9)	0.4
EuroSCORE II (%)	0.89 (0.68–1.47)	1.25 (0.91–1.62)	0.04
Hypertension, *n*	32 (97%)	23 (100%)	>0.9
Diabetes mellitus, *n*	10 (29%)	7 (30%)	>0.9
COPD, *n*	7 (21%)	2 (9%)	0.3
PVD, *n*	6 (18%)	8 (35%)	0.2
Renal insufficiency, *n*	1 (3%)	2 (9%)	0.6
LV EF (%)	61 ± 5.7	61 ± 7.5	>0.9
RV FAC (%)	47.2 ± 10.5	45.1 ± 10.9	0.5
Use of CPB, *n* (%)	4 (12%)	2 (9%)	>0.9
Norepinephrine support, *n*	19 (56%)	16 (70%)	0.6
Norepinephrine dose (µg/kg/min)	0.02 (0–0.07)	0.04 (0.01–0.08)	0.3
Mechanical ventilation parameters			
Tidal volume (mL)	530 (500–568)	520 (500–560)	0.7
Respiratory rate (breaths/minute)	12 (12–12)	12 (12–12)	0.3
PEEP (cm H_2_O)	5 (5–5)	5 (5–5)	0.6
Peak inspiratory pressure (cm H_2_O)	17 (16–20)	17 (15.5–20)	>0.9
Static compliance (mL/cm H_2_O)	56 ± 15	52 ± 10	0.2

COPD, chronic obstructive pulmonary disease; PVD, peripheral vascular disease; LV EF, ejection fraction of the left ventricle; RV FAC, fractional area change of the right ventricle; CPB, cardio-pulmonary bypass; IBW, ideal body weight; PEEP, positive end-expiratory pressure. Values are expressed as mean ± SD or median [interquartile range], as appropriate.

**Table 2 jcm-12-02569-t002:** Hemodynamic parameters before and after the End-Inspiratory Occlusion Test (baseline 1, EIO), before and after the End-Expiratory Occlusion Test (baseline 2, EEO) and before and after a fluid challenge (baseline 3, FC).

	Baseline 1	EIO	Baseline 2	EEO	Baseline 3	FC
**HR** (min^−1^)						
responders	74.2 ± 13.6	74.6 ± 13.9	74.2 ± 14	73.7 ± 14	74.6 ± 13.6	72.4 ± 12.9
non-responders	74 ± 13.4	73.9 ± 13.9	73.8 ± 13.5	73.4 ± 13.4	74.3 ± 13.5	72.5 ± 13.5
**MAP** (mmHg)						
responders	74.6 ± 8.5	71.4 ± 9.5	74.7 ± 8.7	76.1 ± 9.7	75.6 ± 7.6	82.2 ± 11.8
non-responders	73.5 ± 8.6	71.8 ± 10.5	72.4 ± 8	74 ± 8.4	72.8 ± 7.4	78.4 ± 11.5
**CVP** (mmHg)						
responders	5.4 ± 2.4	6.4 ± 2.6	5.3 ± 2.2	4.6 ± 2.3	5.4 ± 2.2	7.2 ± 2.3
non-responders	6.4 ± 2.4	7.3 ± 3	6.2 ± 2.6	6.2 ± 3.1	6.3 ± 2.7	7.9 ± 2.8
**CI** (L/min/m^2^)						
responders	2.5 ± 0.5	2.2 ± 0.6	2.5 ± 0.5	2.6 ± 0.6	2.4 ± 0.6	3.1 ± 0.7
non-responders	2.8 ± 0.7	2.5 ± 0.7	2.8 ± 0.7	2.9 ± 0.8	2.9 ± 0.8 ^a^	3.1 ± 0.8
**SV** (mL)						
responders	66.9 ± 16.6	58.4 ± 16.3	66.7 ± 16.4	71.4 ± 15.9	66.7 ± 16.7	84.7 ± 16.9
non-responders	76.6 ± 25.7	69.9 ± 26.5	75.7 ± 24.9	80.3 ± 25.5	78.2 ± 26.8	86.2 ± 27.4
**SVV** (%)						
responders	17.1 ± 7.9	16.8 ± 7	17.1 ± 8.3	10.5 ± 6.8	16.6 ± 8.3	7.6 ± 3.1
non-responders	12.9 ± 5.1 ^a^	12.8 ± 6.1 ^a^	12.9 ± 5.5 ^a^	8.2 ± 6.2	12.3 ± 5.1	6.6 ± 2.7

HR, heart rate; MAP, mean arterial pressure; CVP, central venous pressure; CI, cardiac index; SV, stroke volume; SVV, stroke volume variation. Values are expressed as mean ± SD. ^a^ *p* < 0.05, responders vs. non-responders.

**Table 3 jcm-12-02569-t003:** Changes in hemodynamic parameters after respiratory occlusion tests and volume expansion.

	EIO	EEO	EIO + EEO	FC
**ΔHeart rate** (%)				
responders	1 ± 3	−1 ± 2	−1 ± 4	−2 ± 6
non-responders	0 ± 3	−1 ± 2	0 ± 4	−3 ± 6
**ΔMAP** (%)				
responders	−4 ± 7	2 ± 5	3 ± 9	9 ± 13
non-responders	−3 ± 6	2 ± 4	3 ± 7	8 ± 14
**ΔCVP** (%)				
responders	19 ± 26	−16 ± 24	−34 ± 38	36 ± 30
non-responders	14 ± 23	−5 ± 24	−18 ± 36	48 ± 90 ^a^
**ΔCI** (%)				
responders	−12 ± 9	7 ± 4	19 ± 11	26 ± 8
non-responders	−10 ± 8	6 ± 4	16 ± 10	7 ± 4 ^a^
**ΔSV** (%)				
responders	−13 ± 9	8 ± 6	20 ± 12	29 ± 13
non-responders	−10 ± 8	7 ± 4	16 ± 10	11 ± 7 ^a^
**ΔSVV** (%)				
responders	2 ± 19	−31 ± 42	33 ± 40	−50 ± 20
non-responders	0 ± 24	−34 ± 45	34 ± 53	−33 ± 55

EIO, end-inspiratory occlusion test; EEO, end-expiratory occlusion test; FC, fluid challenge; HR, heart rate; MAP, mean arterial pressure; CVP, central venous pressure; CI, cardiac index; SV, stroke volume; SVV, stroke volume variation. Values are expressed as mean ± SD. ^a^ *p* < 0.05, responders vs. non-responders.

**Table 4 jcm-12-02569-t004:** Univariate logistic regression analysis for the dependent parameter ΔCI during fluid challenge, binarized as ≥15% vs. <15%.

Independent Variable	OR (95% Confidence Interval)	AUC	*p*-Value
ΔCI (%) during EIO	0.98 (0.92, 1.04)	0.55	0.5
ΔSV (%) during EIO	0.96 (0.89, 1.02)	0.60	0.2
ΔCI (%) during EEO	1.06 (0.93, 1.12)	0.59	0.4
ΔSV (%) during EEO	1.04 (0.94, 1.17)	0.58	0.4
ΔCI (%) during EIO + EEO	1.03 (0.97, 1.084)	0.59	0.4
ΔSV (%) during EIO + EEO	1.04 (0.99, 1.096)	0.63	0.2
Baseline SVV	1.1 (1.01, 1.23)	0.65	0.05

AUC, area under curve; CI, cardiac index; EEO, end-expiratory occlusion; EIO, end-inspiratory occlusion; OR, odds ratio; SV, stroke volume; SVV, stroke volume variation.

**Table 5 jcm-12-02569-t005:** Comparison of the ability of selected variables to predict fluid responsiveness.

	Threshold	Sensitivity (%)	Specificity (%)	PPV (%)	NPV (%)	ROC AUC
**ΔCI**						
EIO	−8.3	64.7	47.8	55.3	57.5	0.552
EEO	5.3	70.6	60.9	64.3	67.4	0.590
EIO + EEO	16.7	61.8	69.6	67	65	0.593
**ΔSV**						
EIO	−6	73.5	47.8	58.4	64.3	0.598
EEO	7	55.9	65.2	61.6	59.7	0.580
EIO + EEO	15.8	70.6	60.9	64.3	67.4	0.631
**SVV**						
baseline	14	58.8	69.6	65.9	62.8	0.645

PPV, positive predictive value; NPV, negative predictive value; EIO, end-inspiratory occlusion test; EEO, end-expiratory occlusion test; CI, cardiac index; SV, stroke volume; SVV, stroke volume variation.

## Data Availability

Data available on request due to ethical restrictions. Public data availability was not specifically stated in the informed consent obtained from study participants.

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
