# Peer review of "Prediction of Fluid Responsiveness Using Combined End-Expiratory and End-Inspiratory Occlusion Tests in Cardiac Surgical Patients"

_jcm, 2023, doi:10.3390/jcm12072569_

Round 1

Reviewer 1 Report

The article under review represents a study investigating if respiratory occlusion tests can predict fluid responsiveness in cardiac surgical patients with protective ventilation. The authors conclude that the respiratory occlusion tests failed to predict the target. Strengths of the study include a well-written manuscript and a prospective study approach. However, the following points should be addressed: 

·         In the statistics section, the authors write: “The ability of CI change induced by individual respiratory tests or their combination to predict fluid responsiveness was evaluated using logistic regression followed by receiver operating characteristic (ROC) analysis.” Please elaborate further on the results of the logistic regression and what input predictors were used.

·         It would be beneficial if the results from the logistic regression could be presented in a table, showing the predictors and significance.

·         What was the missing rate for the variables analyzed in the study, and were any imputation techniques used?

·         In the discussion, the authors note that “several studies also report a failure of respiratory occlusion tests to predict fluid responsiveness reliably.” Please, further clarify what the current study adds to the body of knowledge in this field.

·         It should be noted in the limitations section that 46 % of the patients who met the inclusion criteria were excluded from the current study.

Reviewer 2 Report

In this article, the authors analyzed the influence of dynamic respiratory maneuvers for minimizing intrathoracic pressure (end-expiratory occlusion) or increasing intrathoracic pressure (end-inspiratory occlusion) to predict fluid responsiveness in cardiac surgery patients.

This study is well carried out, consistent with all those of the same type carried out with the aim of predicting fluid responsiveness with dynamic maneuvers. The originality of this work lies not in the maneuvers carried out but, in the population studied and the haemodynamic monitoring system implemented. However, this study is negative.

One potential explanation that the authors should discuss is why volume expansion was performed in these patients?  Their heart rate are normal, as are their cardiac index and mean arterial blood pressure. Patients were under vasopressor support but only with very low dosages noradrenaline. Therefore, it is likely that in the study population, very few patients were authentically hypovolemic...
